# International Tourists' Loyalty to Ho Chi Minh City Destination—A Mediation Analysis of Perceived Service Quality and Perceived Value

**Khuong Ngoc Mai [1], Phuong Ngoc Duy Nguyen [1,*]**  **and Phuong Thi Minh Nguyen [2]**

[1] School of Business, International University, Vietnam National University Ho Chi Minh City (VNU-HCM), Linh Trung Ward, Thu Duc District, Ho Chi Minh City 700000, Vietnam; mnkhuong@hcmiu.edu.vn
[2] Department of Tourism, Hong Bang International University, 215 Dien Bien Phu, Ward 15, Binh Thanh District, Ho Chi Minh City, Vietnam; phuongntm2@hiu.vn
[*] Correspondence: nndphuong@hcmiu.edu.vn

**Abstract:** At present, tourism is a key component of the modern Vietnamese economy. However, it is increasing the country's pitifully low return visitor rate. The aim of this study was to explore the structural relationships among destination attributes, perceived service quality, perceived value, and loyalty of international tourists in Ho Chi Minh City. The destination attributes were investigated in terms of cultural and historical attractions, local cuisine, perceived price, safety and security, service facilities, natural environment, entertainment and recreation activities, negative attributes, and destination image. Data were collected through a self-administered questionnaire survey of 2073 respondents employed using SmartPLS analysis. The results indicate that destination attributes, perceived service quality, and perceived value have positive and direct influences on tourist loyalty. In addition, the findings also confirmed that these destination attributes indirectly affected tourist loyalty through their perceived value and perceived service quality. Based on the findings, some recommendations and implications are suggested to enhance traveler intention to re-visit and increase awareness of the necessity for sustainable development tourism in Ho Chi Minh City, with their willingness to recommend this place to others.

**Keywords:** destination attributes; perceived service quality; perceived value; tourist loyalty; Vietnam

## 1. Introduction

Tourism has become an indispensable need in cultural and social life worldwide. In economic terms, tourism has become one of the important economic sectors of many countries. The economic benefits brought about by tourism are undeniable, through visitors' consumption of tourism products. The demand of tourists besides the consumption of ordinary goods also has special consumption needs: the need to improve knowledge, sightseeing, health tourism, and relax. Vietnam has become the most attractive destination in the region. The Vietnam General Statistics Office [1] reported that international tourists to the country reached over fifteen million arrivals in 2018, contributing around 9% of the GDP. Especially, Ho Chi Minh (HCM) City attracts the most international tourists, and a city has many advantages for the development of sustainable tourism, because it has many scenic spots, culturally and historically bears the impressions of the Vietnamese, Indians, Chinese, Cham, and Khmer, and also has the influences of French and American cultures. This is reflected through constructions like Reunification Palace, Nha Rong Wharf, the Grand Theater, the Post Office, City Hall, and Ben Thanh Market. In the system of ancient pagodas and churches like Giac Lam, Notre Dame, etc. along the Saigon River from Bach Dang Wharf to the downtown, the international tourists can

travel in a cruise to enjoy nature, visit traditional craft villages, orchards, ornamental plant gardens, floating markets on the river, and Can Gio Ecotourism Resort, inscribed by UNESCO as Vietnam's first mangrove forest biosphere reserve. The city is also a gateway where international tourists can come to many famous tourist destinations including Nam Cat Tien National Park, Mui Ne Cape, and the Binh Chau hot spring.

Although the growth rate of international tourist visitors to Vietnam decreased because of objective issues related to politics and did not reach the target of the central policy, the total revenue of the Vietnam tourism industry still reached its target as shown in Figure 1 [2].

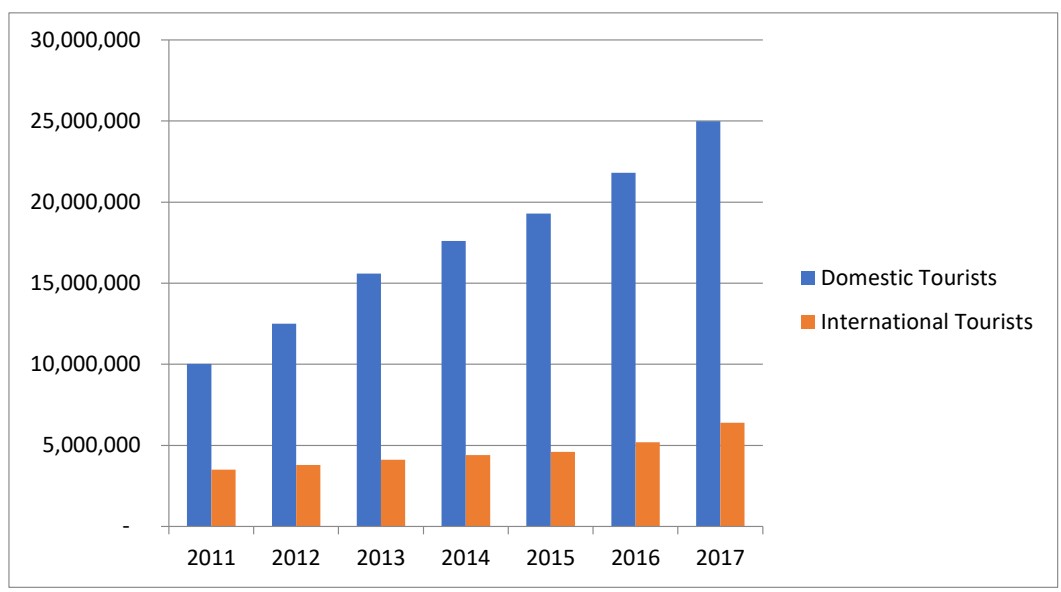

**Figure 1.** Domestic and international tourist arrivals *(Source: HCMC Department Tourism, 2018).*

In the strategy for Vietnam tourism development to 2030, tourism is a key economic sector with high competitiveness. Vietnam has become a particularly attractive destination and is part of the leading tourism development hub in Southeast Asia [3]. By 2050, tourism will be the driving force of the economy. Vietnam has become a prominent global destination, belonging to the leading tourism development group in the Asia-Pacific region. Total revenue from tourists in 2050 will increase by 3.5 to 4 times compared to the projected figures for 2030 [4,5]. To achieve these targets, attracting and retaining international tourist is a priority. Furthermore, in tourism destination management, enhancing tourist loyalty is crucial. In the tourism literature, the strong interrelationship between perceived service quality, perceived value, and tourist loyalty has been proven [4,5]. It is expected to maintain a high level of these factors and making sure that international tourists have many meaningful experiences during their stay in Ho Chi Minh City, raise their awareness about sustainability problems, and campaign for sustainable tourism training in Ho Chi Minh City.

The objective of this study was to identify which destination attributes directly and indirectly affect tourist loyalty mediated by perceived service quality and perceived value. Some empirical evidence has described the impact of destination attributes on tourist loyalty [6–10]. However, these studies focused on the positive relationships of destination attributes and tourist loyalty [8–10]. There is a shortage of studies seeking to explore the negative attributes of a destination and its negative impact on tourist behavior. Furthermore, the relevant literature still lacks investigations of how destination attributes influence tourist loyalty by studying perceived service quality and perceived value as the mediating variables in the relationship. Because of these limits, it is essential to explore the structural relationship among these constructs. This is expected to provide practical evidence to tourism policy makers in order to help them in creating effective strategies for retaining tourist loyalty.

## 2. Literature Review

### 2.1. Tourist Loyalty

The important goal of every tourist destination is to make tourists loyal. The concept of tourist loyalty is measured by a traveler's intention to re-visit a specific destination and recommend the place to others [7,9]. According to Kim [11], a destination is viewed as a special product which consists of culture, artificial attractions, and landscape resources, and tourists' expectation to revisit a place is much lower than other types of product, regardless of the destination. In another word, visitors seem to seek new destinations due to their desire to experience new places [12]. Additionally, Ozturk and Qu [13] argued that regardless of the tourism package, a visitor may not purchase it. As such, a positive recommendation to others is also considered a relevant measurement to predict tourist loyalty. In the same vein, when tourists have a positive memory in a certain destination, they tend to revisit the place and pass on positive word-of-mouth (WOM) to their friends and relatives as well. Similarly, there in increasing use of WOM advertising, since it is the easiest channel and is reliable to access potential visitors [7,8].

In terms of management, tourist loyalty is the critical component for the success and long-term growth of business. However, according to MCST [2], more than 80% of foreigners said they would not visit Vietnam again. This rate is very high compared to our neighbor countries such as Thailand and Malaysia. It is known that customer loyalty not only guarantees revenue, market share development, and positive word-of-mouth, but also reduces the overhead of operations and destination marketing costs [14–16]. Therefore, it is important to manage tourist loyalty as one of the fundamental issues.

### 2.2. Perceived Value

According to Zeithaml [17], perceived value is defined as customer's perceived worth of product or service in relation to the total cost that customers pay for it—that is, tourists' perceptions of what they receive from the destination and what they are given for the attainment of that destination. Likewise, Petrick and Backman [18], perceived value in tourism services is based on tourists' perception of service quality, with financial and non-financial cost perceptions as the determinants. In the study of Rodríguez-Diaz et al. [19], perceived value was examined based on the value for money. A great deal of empirical evidence has confirmed the positive relationship between tourists' perception of value and loyalty [20–24]. When visitors assess high perceived value toward the destination, they tend to increase revisit rates and recommend the destination to others.

### 2.3. Perceived Service Quality

Quality is recognized in terms of two aspects: the process of offering a service and the results of them service [4]. In tourism, the process is examined by the reliability, efficiency, goodwill, and employee competency. Meanwhile, the results of services can be investigated in housing, food, and leisure time facilities [4]. Service quality has been considered as the match between a customer's expectations before using the service and their perceptions after using the service [25]. Perceived service quality is normally described as the overall result of the customer's expectation and the service they received [25]. In other words, when a traveler perceives the service to be excellent compared to their desired level, they tend to strengthen their loyalty toward the service [25]. This depends on the difference between their expectation and the perceived service. Several authors [10,26–29] indicated that perceived service quality is a critical factor to predict customer fulfillment, intention to return, and likelihood of recommending the destination to others. When a traveler has a higher level of perceived service quality, they are willing to experience more, which leads to them revisiting the destination. In addition, it can be assumed that higher perceived service quality has a positive relationship on visitor loyalty [5,30–32].

## 2.4. Destination Attributes

In the tourism and hospitality literature, various destination factors have been recognized, such as physical, psychological, and tangible factors to measure customer perceptions, satisfaction, and resulting traveler loyalty. In the study of Coban [33], satisfaction attributes were classified by tourist attractions, cultural attractions, natural environment, basic facilities, substructures, and access possibilities. Similarly, Supitchayangkool [34] studied the structural relationship between service quality, satisfaction, and tourists' intention to revisit Pattaya, Thailand based on various destination dimensions such as fairness of price, hygiene, amenities, value for money, logistics, food, and security. Looking at different aspects, Rajesh [35] studied the influence of physical factors (natural attractions, historical and cultural attractions, entertainments and infrastructure), and psychological factors (destination affordability, travel environment) on tourist perception, and their satisfaction and destination loyalty. In Vietnam particularly, there is little study on international tourist loyalty. Ho and Tran [36] investigated the impact of environment, facilities, culture and society, entertainment activities, and cuisine on international tourists' revisit intention and likelihood of recommending Nha Trang City to others.

In this research, tourist loyalty was analyzed and predicted by nine destination attributes:

The first one, the factor of historical and cultural attractions (CULHISAT) is alleged as a key economic driver which policy makers attempt to exploit by promoting cultural activity in the country. As a result, there is increasing attention paid to the interrelation between culture and travel [37]. Cultural tourism consists of some typical activities such as participating in cultural festivals, visiting cultural and historical or themed sites such as museums, theaters, exhibitions, religious centers, etc. Cultural and historical attraction is the prerequisite element for the formation and development of a region/country's tourism. It is the fundamental element of travel impetus, and impacts a traveler's destination decision. There is some empirical evidence indicating that different culture experiences are one of the destination preferences that were significantly related to tourist loyalty [7]. These culture experiences create the dimensions of tourism satisfaction [38].

The second attribute, local cuisine (LOCUIS) has also proved to be an important element that leads to traveler destination decisions [39]. They tend to seek different local cuisines which are not available in their home country [40]. Moreover, local cuisine is a vital element for enlightening tourists [41]. Previous studies [42–44] proved that local dishes were key determinants of tourist positive perception, satisfaction, and revisit intention. When tourists feel satisfied with the local cuisine, they tend to visit the place again and recommend this destination to others.

Then, perceived price (PERPRI) refers to the status that a customer sacrifices to acquire the service [14]. It consists of both monetary and non-financial costs in customers' perception of what they received and what they paid. Non-financial costs are regarded as wasted time, convenience, search costs, and brand image. Budget and travel costs are the most prerequisite element that travelers prioritize when they decide to travel. Ozturk and Qu found that some tourists can remember the exact price of a past trip, while others may recall expensive or inexpensive fees of a past vacation. It is therefore the perceived financial and non-financial factors that generate the overall perceived sacrifice in a tourist's perception of the destination [18]. Visitors tend to get higher satisfaction with affordable destinations, and these satisfied customers are more likely to revisit this destination and give a positive recommendation to others.

The fourth attribute, safety and security (SAFESECU), is also a major concern of many tourists when they select a destination to travel. Hence, a tourist's perception of safety and security is a critical factor in a traveler's decision to visit a destination [45]. According to Tarlow [46], the tourist considers the terrorism threat in the country, accommodation facilities, transportation, police service, common criminality, and violence. Countries with a poor image for safe tourism may lose the competitive advantage to attract international tourists, while countries with a safe travel image will attract more customers. The influence of safety and security on tourist satisfaction and loyalty was also considered as a topic interest for many researchers. For example, Chi [8] found that safety led to higher customer

satisfaction. In another study, security and safety had a significant and direct impact on tourist satisfaction and tourist loyalty [47].

The fifth attribute, service facilities (INFRAS), are tangible attributes such as tourist sites, major buildings, and popular places which offer customers access to facilities in that destination. They include accommodation units, food and beverage services, transportation, telecommunications, shopping malls, sport centers, travel agencies, information centers, etc. These facilities also appear as key factors in travelers' destination selection [48]. It could be argued that a tourist will revisit a destination when they are pleased with the quality of the facilities and services [49]. Onditi [50] also indicated the role of ddestination facilities, including all related activities such as restaurants, gift shops, and tour guides. Moreover, throughout destination facilities, service quality directly impacted tourist satisfaction and loyalty [51].

Natural environment (NATENVI) is another key determinant of a tourist's destination selection. In this industry, the natural environment consists of the weather, beaches, lakes, mountains, deserts [52], etc., and these elements lead to the favorable attraction of a tourist to the destination. Coban [33] examined the influence of tourists' perception of environmental elements on positive image and tourist pleasure. Authors in Beerli and Martin [52] suggested that tourists' entire environment perception was influenced by the natural ecosystem, which greatly determines tourist loyalty. Some empirical evidence has proved that environmental aspects have a significant relationship with a tourist's perception and revisit intentions [41,53,54].

Besides the discussed factors, entertainment and recreation activities (ENRECACT) are also key determinants. Entertainment and recreation activities consist of outdoor activities, adventure activities, nightlife, shopping, food and beverage activities, etc. [55]. In the tourism literature, entertainment and recreation activities were the main aspects in studying tourists' satisfaction as well as revisit and recommend intention [8,35,36,52].

In the study of Truong and Foster [56], negative destination attributes (NEGAT) reflect the traveler's unfavorable perception about the destination, while positive attributes refer to the opposite. They are both important in evaluating the holiday experiences of visitors in a specific destination [29]. Some empirical evidence indicated that the negative attributes had a great effect on the tourist satisfaction assessment and revisit intention [57]. It can be argued that traveler may recall negative impressions for longer than things that pleased them. Hence, it will be more accurate to evaluate the influence of negative attributes on tourist perceived value, service quality, and return intention.

Finally, the destination image (DESIMA) has received a great deal of attention, and not only in the tourism sector. It is also widely accepted in many market segments [58]. Destination image refers to the impressions or belief a traveler has toward a place [59]. This comprises two elements: cognitive and affective image [59]. The former conveys the belief that tourist holds of a destination. The latter reveals their personal emotions about the destination. If a traveler has a good travel experience, then they tend to have a positive image of the destination. As a result, the tourist will have an overall positive assessment of the destination. Several prior researchers have already studied the impact of destination image on the tourist destination selection process, their perceived value [13,22], perceived quality [60], and their loyalty toward this destination [7,8,60–65].

## 3. Conceptual Framework

From the above theoretical background, previous related theories, and studies, we propose a conceptual framework to investigate the direct and indirect influences of destination attributes on tourist loyalty, as well as the causal relationships among perceived service quality, perceived value, and tourist loyalty. Figure 2 illustrates the hypothetical causal model of this study, which was applied from previous hypothesized models.

**Hypothesis 1 (H1):** *There are nine destination attributes (culhisat, desima, enrecact, infras, locuis, natenvi, negat, perpri, and safesecu) that affect perceived value.*

**Hypothesis 2 (H2):** *There are nine destination attributes (culhisat, desima, enrecact, infras, locuis, natenvi, negat, perpri, and safesecu) that affect perceived service quality.*

**Hypothesis 3 (H3):** *There are nine destination attributes (culhisat, desima, enrecact, infras, locuis, natenvi, negat, perpri, and safesecu) that affect tourist loyalty.*

**Hypothesis 4 (H4):** *The effect of nine destination attributes (culhisat, desima, enrecact, infras, locuis, natenvi, negat, perpri, and safesecu) on tourist loyalty is mediated by perceived value.*

**Hypothesis 5 (H5):** *The effect of nine destination attributes (culhisat, desima, enrecact, infras, locuis, natenvi, negat, perpri, and safesecu) on tourist loyalty is mediated by perceived service quality.*

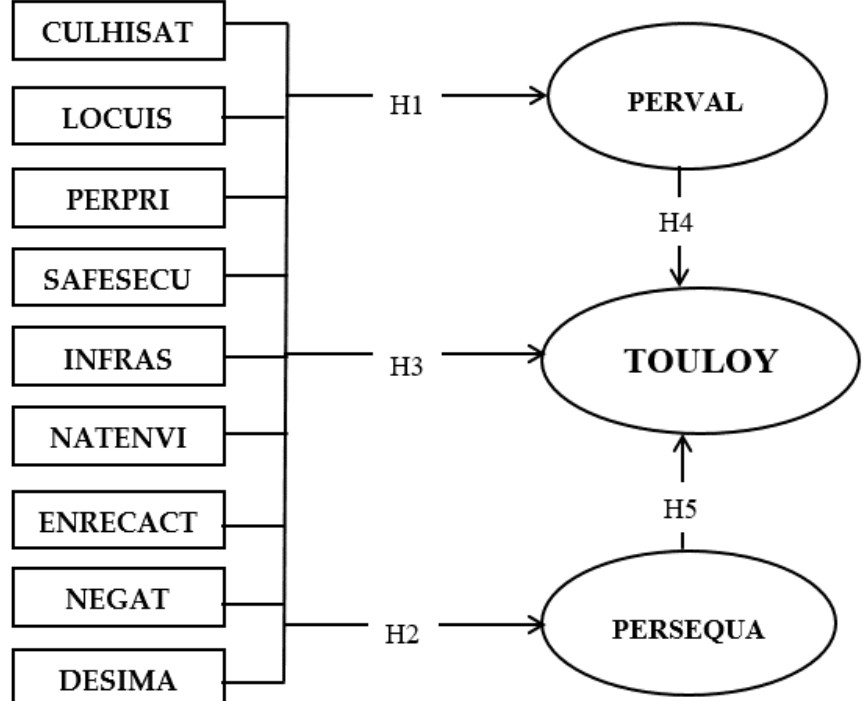

**Figure 2.** Proposed conceptual framework. Culhisat: historical and cultural attractions; locuis: local cuisine; perpri: perceived price; safesecu: safety and security; infras: service facilities; natenvi: natural environment; enrecact: entertainment and recreation activities; negat: negative destination attributes; desima: destination image; perval: perceived value; touloy: tourist loyalty; persequa: perceived quality.

## 4. Methodology

### 4.1. Measurement of Constructs

The scale measurements for this research were developed in self-administered questionnaires. These statements were measured on a 5-point scale ranging from "strongly disagree" to "strongly agree". The scales for destination attributes were adapted from Chi [66], Tran [67], Beerli and Martin [52], and Vinayek et al. [68]. The scale for perceived value was derived from Chen [69]; that for perceived service quality was from Fornell et al. [70]; and finally, tourist loyalty scale was adapted from Oliver [71] and Taylor [72]. Some demographic variables to describe the tourist information such as age, gender, nationality, and length of stay of respondents were also evaluated.

### 4.2. Data Collection and Sample

To ensure a good sample size, we chose Ho Chi Minh City as the subject, as it is the largest city and center of financial and cultural areas in South Vietnam. Besides, the city converges all cuisines of

Vietnam from North to South. It has a wide range of historical cultural sites such as Dragon Wharf and the Museum of Vietnamese History. Regarding recreational areas, there are several around the city where visitors can enjoy different activities such as cooking classes, motorcycle tours, Mekong Delta day cruises, Saigon River cruises, and visits to riverside markets and local villages. The monastery system has become an important part of travelling in HCMC, with many temples and pagodas like the Jade Emperor Pagoda and Floating Temple, where the visitors can learn Vietnamese religions and appreciate these wonderful and holy architectural sites. It also has a convenient location close to many beautiful beaches like Vung Tau beach, Mui Ne Cape, Mekong Delta, and Can Gio Mangrove forest that have potential for eco-tourism thanks to biosphere reserves as shown in Figure 3. below. Ho Chi Minh draws the highest number of international tourists to Vietnam, and at this time it is voted as the country's second most attractive travel destination by the world largest travel site [2].

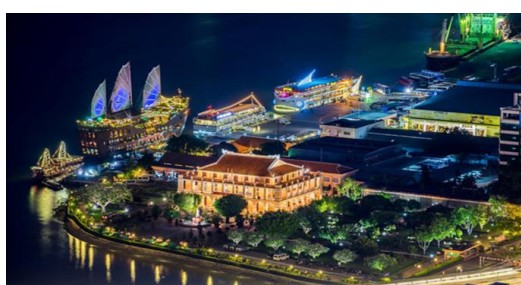

(**a**) Nha Rong Wharf (2018). By: Xuan Nguyen.

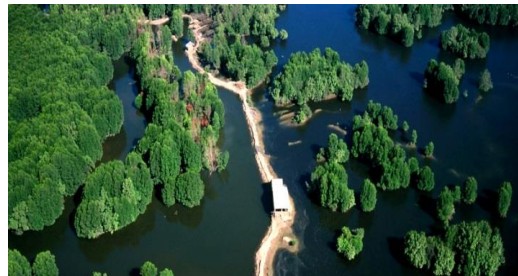

(**b**) Can Gio Mangrove forest.

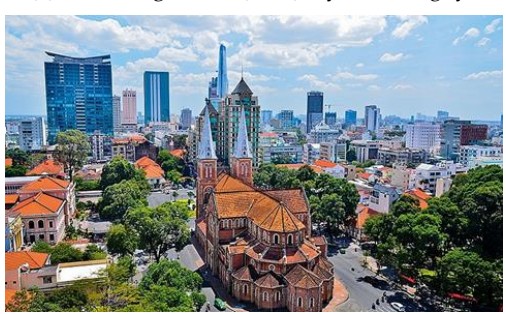

(**c**) Notre Dame Cathedral in Ho Chi Minh City.

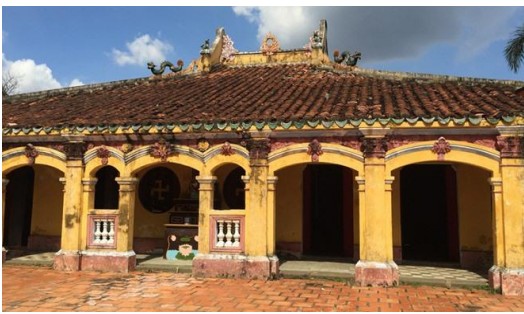

(**d**) Architecture: 160-year-old Giac Vien Pagoda.

**Figure 3.** Some typical tourists' attractive places in Ho Chi Minh City as.

The authors chose a sample size: variable ratio of 10:1 (which is considered very well). With a total of 75 observed variables, this study needed a sample size of 750 respondents. The survey was delivered directly to international tourists at the airport and best tourist attractions in Ho Chi Minh, like Binh Tay Market, War Remnants Museum, Cu Chi Tunnels, Reunification Palace, Ho Chi Minh Central Post Office, Notre Dame Cathedral, and Bitexco Tower and Sky Deck, with directions and accurate contents to help them answer correctly. Surveys were collected from mid-2015 to mid-2016 in order to reach the required number of samples. Finally, there were 2073 responses providing relevant and sufficient answers, resulting in an effective response rate.

According to the descriptive analysis, the results indicate that the survey attracted a nearly equal number of males (53.4%) and females (46.6%). Guests who were asked possessed a diverse variety of nationalities, such as 31.6% Europeans and 46.4% Asians. Additionally, there was a large portion of guests who had earned a Bachelor's degree (38.6%). Following was 21.1% for guests with Master's or degrees or higher. Most of the guests had been to Ho Chi Minh City only once (64.9%), and did so for leisure activities (62.9%).

*4.3. Analysis*

The partial least squares (PLS) technique was applied in this study. Hair et al. [73] suggest that PLS be used in exploring the prediction of key predictors and investigating the structural

complex relationships among constructs. Data analysis followed a two-step approach [74]. First, the measurement model was applied to evaluate the construct's reliability and validity by testing the factor loadings, Cronbach's alpha, and composite reliability [75–77]. Second, the hypotheses were tested to examine the complex relationships among constructs, and then finally the research hypotheses were tested.

## 5. Results

### 5.1. Measurement Model

Table 1 depicts the analysis of the measurement model by assessing the reliability and validity of the constructs. The results of factor loadings were all above 0.7 [73–77]. Composite reliabilities (CRs) were acceptable, ranging from 0.837 to 0.885. The average variance extracted (AVE) was also above the threshold of 0.5. Thus, the convergent validity and reliability of constructs were confirmed [74].

**Table 1.** Measurement model evaluation.

| Constructs | Items | Factor Loadings | CR | AVE |
|---|---|---|---|---|
| Cultural and Historic Attractions (CULHISAT) | Variety of historic sites/museums | 0.845 | 0.885 | 0.658 |
| | Variety of religious sites/museums | 0.832 | | |
| | Reasonable price for sightseeing | 0.764 | | |
| Destination Image (DESIMA) | Wide selection of restaurants/cuisine | 0.707 | 0.837 | 0.562 |
| | Wide variety of shop facilities | 0.771 | | |
| | Wide variety of relaxing activities | 0.799 | | |
| | Regional transportation hub | 0.717 | | |
| Entertainment (ENRECACT) | Variety of special events/festivals | 0.748 | 0.821 | 0.605 |
| | Variety of spa/massage/healing options | 0.810 | | |
| | Variety of entertainment (music, bars, pubs, coffee shops, etc.) | 0.775 | | |
| Infrastructure (INFRAS) | Quality and cleanliness of lodging facilities | 0.754 | 0.855 | 0.664 |
| | Service in lodging facilities | 0.845 | | |
| | Availability of travel information | 0.842 | | |
| Local Cuisine (LOCUIS) | Variety of cuisine (Vietnamese and foreign food) | 0.704 | 0.866 | 0.619 |
| | Quality of food | 0.807 | | |
| | Convenience of meals | 0.842 | | |
| | Service in restaurants | 0.789 | | |
| Natural Environment (NATENVI) | Cleanliness | 0.730 | 0.851 | 0.588 |
| | Air and noise pollution | 0.786 | | |
| | Language barriers | 0.773 | | |
| | Friendly local people | 0.779 | | |
| Negative Attributes (NEGAT) | Crowded and dangerous traffic | 0.877 | 0.868 | 0.766 |
| | Many beggars and street vendors | 0.874 | | |
| Perceived Price (PERPRI) | Happy with the price | 0.730 | 0.837 | 0.562 |
| | Satisfied with this price | 0.746 | | |
| | This is the price that I would expect to pay | 0.766 | | |
| | I am pleasantly surprised with the price | 0.758 | | |
| Perceived Quality (PERSEQUAL) | My perception of the overall quality of HCMC is good | 0.730 | 0.878 | 0.591 |
| | My perception of the product quality of HCMC is good | 0.797 | | |
| | My perception of the service quality of HCMC is good | 0.762 | | |
| Perceived Value (PERVA) | The experience in HCMC was good for the price I paid | 0.744 | 0.867 | 0.622 |
| | My total expenditures were reasonable for the trip I experienced | 0.847 | | |
| | The experience in HCMC was good for the time I spent | 0.833 | | |
| | My total time spent was reasonable for the trip I experienced | 0.722 | | |
| Safety and Security (SAFESECU) | Political stability | 0.824 | 0.853 | 0.659 |
| | Tourists protected by law and order | 0.837 | | |
| | Low crime rate | 0.773 | | |
| Tourist Loyalty (TOLOY) | Intention to revisit HCMC | 0.863 | 0.871 | 0.693 |
| | Willingness to recommend HCMC as a favorable destination to others | 0.766 | | |
| | Intention to visit again more times | 0.865 | | |

CR: composite reliability; AVE: average variance extracted; HCMC: Ho Chi Minh City.

The discrimination validity was also assessed through indicator cross-loadings and intercorrelations between constructs [73–77]. The former indicated that all latent constructs were higher than their loadings with all other constructs [73]. The latter one showed the indicators of the intercorrelations among the latent constructs by the square root of the AVE of a single construct reflected in bold diagonals [74]. The results in Table 2. confirmed the qualified requirement for assessing the discriminant validity.

**Table 2.** Discriminant validity

|  | **A** | **B** | **C** | **D** | **E** | **F** | **G** | **H** | **I** | **J** | **K** | **L** |
|---|---|---|---|---|---|---|---|---|---|---|---|---|
| A | **0.811** | | | | | | | | | | | |
| B | 0.304 | **0.750** | | | | | | | | | | |
| C | 0.382 | 0.444 | **0.778** | | | | | | | | | |
| D | 0.326 | 0.467 | 0.415 | **0.815** | | | | | | | | |
| E | 0.342 | 0.492 | 0.374 | 0.494 | **0.787** | | | | | | | |
| F | 0.536 | 0.177 | 0.332 | 0.369 | 0.233 | **0.767** | | | | | | |
| G | -0.036 | 0.066 | 0.045 | -0.001 | 0.090 | -0.109 | **0.875** | | | | | |
| H | 0.364 | 0.517 | 0.467 | 0.532 | 0.588 | 0.281 | -0.022 | **0.750** | | | | |
| I | 0.317 | 0.498 | 0.390 | 0.571 | 0.539 | 0.305 | -0.076 | 0.512 | **0.769** | | | |
| J | 0.342 | 0.515 | 0.409 | 0.433 | 0.498 | 0.309 | -0.009 | 0.471 | 0.551 | **0.788** | | |
| K | 0.290 | 0.267 | 0.259 | 0.395 | 0.345 | 0.342 | -0.005 | 0.334 | 0.375 | 0.306 | **0.812** | |
| L | 0.277 | 0.286 | 0.383 | 0.301 | 0.284 | 0.280 | 0.065 | 0.308 | 0.363 | 0.460 | 0.250 | **0.833** |

Note: A: CULHISAT; B: DESIMA; C: INFRAS; D: LOCUIS; E: NATENVI; F: NEGAT; G: PERPRI; I: PERSEQUA; J: PERVA; K: SAFESECU; L: TOLOY. Diagonals (in bold) represent square root of the AVE.

## 5.2. Assessing Structural Model Results

According to Hair et al. [75], the variance inflation factor (VIF) was applied to check the multicollinearity—according to authors in [75,76], if the VIF is greater than 5, it would be reached. Results showed that all the estimated values of VIF were smaller than 5, ranging from 1.279 to 2.093. Therefore, the collinearity did not negatively influence the predictor variable in the structural model. Thus, the analysis had no issues. PLS uses the nonparametric bootstrapping proposed by Esmaeilifar et al. [74] to calculate the significant coefficient. After the bootstrapping process, *t*-values were calculated to see if the coefficients were considerably different from zero. According to Hair et al. [73] and Henseler et al. [78], the techniques of non-parametric bootstrapping were utilized in every single sign change—2073 cases and 1000 micro-samples.

Hypothesis 1 was tested, and the results showed that tourists' perceived value (PERVAL) was mainly affected by important factors at the 95% confidence level: CULHISAT ($\beta$ = 0.040, $p$ = 0.038), DESIMA ($\beta$ = 0.263, $p$ = 0.000), ENRECACT ($\beta$ = 0.096, $p$ = 0.000), INFRAS ($\beta$ = 0.059, $p$ = 0.000), LOCUIS ($\beta$ = 0.209, $p$ = 0.000), NATENVI ($\beta$ = 0.098, $p$ = 0.000), NEGAT ($\beta$ = -0.035, $p$ < 0.010), PERPRI ($\beta$ = 0.080, $p$ = 0.002), and SAFESECU ($\beta$ = 0.044, $p$ = 0.037). The results indicated that destination image had the highest impact on tourists' perceived value. Furthermore, the model could explain 39.6% of the variance of perceived value.

The results of Hypothesis 2 showed that tourists' perceived service quality (PERSEQUA) was largely influenced by the biggest to the smallest predictors: INFRAS ($\beta$ = 0.226, $p$ = 0.000), LOCUIS ($\beta$ = 0.225, $p$ = 0.000), DESIMA ($\beta$ = 0.176, $p$ = 0.000), NEGAT ($\beta$ = -0.103, $p$ = 0.000), SAFESECU ($\beta$ = 0.092, $p$ = 0.000), PERPRI ($\beta$ = 0.083, $p$ = 0.002), ENRECACT ($\beta$ = 0.047, $p$ = 0.031), and NATENVI ($\beta$ = 0.044, $p$ = 0.042). However, CULHISAT had no impact on perceived service quality. The adjusted R squared value was 0.472, which means 47.2% of the variation in perceived service quality could be explained by these destination attributes.

The findings of Hypothesis 3 indicate that tourist loyalty (TOULOY) was directly influenced by ENRECACT ($\beta$ = 0.183, $p$ = 0.000), NATENVI ($\beta$ = 0.076, $p$ = 0.004), NEGAT ($\beta$ = 0.085, $p$ < 0.000), SAFESECU ($\beta$ = 0.052, $p$ < 0.028), PERSEQUA ($\beta$ = 0.108, $p$ = 0.000), and PERVA ($\beta$ = 0.308, $p$ = 0.000). From the results, five destination attributes had no relationship with tourists' loyalty: cultural and

historical attractions, destination image, infrastructure, local cuisine, and perceived price. Furthermore, the tourists' loyalty model could explain 27.9% of the variation of the construct.

As mentioned, perceived value was positively affected by CULHISAT (β = 0.040, *p* = 0.038), DESIMA (β = 0.263, *p* = 0.000), ENRECACT (β = 0.096, *p* = 0.000), INFRAS (β = 0.059, *p* = 0.000), LOCUIS (β = 0.209, *p* = 0.000), NATENVI (β = 0.098, *p* = 0.000), NEGAT (β = -0.035, *p* < 0.010), PERPRI (β = 0.080, *p* = 0.002), and SAFESECU (β = 0.044, *p* = 0.037). These destination factors directly influenced the mediate variable of perceived value (H1) and then perceived value directly caused an effect on tourist loyalty with β = 0.308, *p* = 0.000 (H4 in Table 3). From the findings, by examining the mediation of perceived value, the destination attributes cultural and historical attractions, local cuisine, natural environment, infrastructure, destination image, negative attributes, perceived price, and safety and security showed indirect effects on tourist loyalty. Therefore, it can be concluded that improvements to these destination attributes would cause higher tourist loyalty by enhancing perceived value.

**Table 3.** Moderating effects of perceived value and perceived service quality.

| Hypothesis | Relationships | Path Coefficients | Decisions |
|---|---|---|---|
| – | PERVAL → TOULOY | 0.308 *** | – |
| H4a | CULHISAT → PERVA → TOLOY | 0.012 ** | Supported |
| H4b | DESIMA → PERVA → TOLOY | 0.081 *** | Supported |
| H4c | ENRECACT → PERVA → TOLOY | 0.029 *** | Supported |
| H4d | INFRAS → PERVA → TOLOY | 0.018 ** | Supported |
| H4e | LOCUIS → PERVA → TOLOY | 0.064 *** | Supported |
| H4f | NATENVI -> PERVA -> TOLOY | 0.030 *** | Supported |
| H4g | NEGAT → PERVA → TOLOY | −0.011 ** | Supported |
| H4h | PERPRI → PERVA → TOLOY | 0.024 ** | Supported |
| H4i | SAFESECU → PERVA → TOLOY | 0.013 ** | Supported |
| | PERSEQUA → TOULOY | 0.108 *** | – |
| H5a | CULHISAT → PERSEQUA → TOLOY | 0.003 | Not supported |
| H5b | DESIMA → PERSEQUA → TOLOY | 0.019 *** | Supported |
| H5c | ENRECACT → PERSEQUA → TOLOY | 0.005 | Not supported |
| H5d | INFRAS → PERSEQUA → TOLOY | 0.028 *** | Supported |
| H5e | LOCUIS → PERSEQUA → TOLOY | 0.024 *** | Supported |
| H5f | NATENVI → PERSEQUA → TOLOY | 0.005 | Not supported |
| H5g | NEGAT → PERSEQUA → TOLOY | −0.011 *** | Supported |
| H5h | PERPRI → PERSEQUA → TOLOY | 0.009 ** | Supported |
| H5i | SAFESECU → PERSEQUA → TOLOY | 0.010 ** | Supported |

** *p* < 0.05, *** *p* < 0.001 (one–tailed).

From the hypothesis (H5) test results, it was concluded that infrastructure, local cuisine, destination image, negative attributes, safety and security, perceived price, natural environment, entertainment, and recreation activities directly and positively influenced tourists' perceived service quality. This means that through the intervening variable of perceived service quality (β = 0.108, *p* = 0.000) on tourist loyalty, these predictors of destination attributes created an indirect effect on tourist loyalty.

## 6. Discussions

According to the descriptive analysis, most of the guests had traveled for the first time to Ho Chi Minh City (about 64.9%), and this visit was for leisure activities in 62.9% of cases, accounting for the biggest percentage of those visiting HCM City. This indicates that HCM City is a potential area for the development of sustainable tourism, and the government should invest more in travel to make international tourists return to HCM City, such as establishing an organization for sustainable development tourism and monitoring the sustainable development of city tourism.

Our findings support the assumptions of the study in predicting international tourist destination loyalty to Ho Chi Minh City. The results indicated that cultural and historical attractions, destination

image, natural environment, and entertainment and recreation activities had direct and positive effects on tourist loyalty, while infrastructure, local cuisine, and perceived price had no impact on tourist loyalty. Moreover, the mediate variables of perceived value and perceived service quality also had direct and indirect influences on tourist loyalty. These results demonstrated that international travelers had higher tourist loyalty toward Ho Chi Minh City, Vietnam when they had higher perceived values of Vietnam's local cuisine, perceived price, cultural and historical attractions, natural environment, and entertainment and recreation activities.

These findings are supported by some previous research [7,8,35,36,41–44,52–54]. They agreed that destination attributes including local cuisine, perceived price, natural environment, cultural and historical attractions, and entertainment and recreation activities positively affected tourist intention to revisit this destination as well as recommend it to others.

However, a significant difference was found in the mediate variable of perceived service quality. Several studies [10,26–29] found that perceived service quality was an important and effective factor to predict customer satisfaction, intention to return, and willingness to recommend to others. Nevertheless, in this study, this relationship was not accepted in the case of Ho Chi Minh City. As a result, the direct effects of infrastructure, local cuisine, and perceived price on perceived service quality did not create any impact on tourist loyalty.

In sum, the results confirmed the complex relationship between destination attributes, perceived value, perceived service quality, and tourist loyalty presented in Table 3. The majority of our hypotheses were thus supported. From the findings, some implications and recommendations have been suggested for the tourism sector, destination marketers, and managers in Ho Chi Minh City specifically, and in Vietnam in general.

Firstly, this study provides practical evidence on the complex relationships between the perceived value and loyalty of foreign tourists in HCM City, as well as the impact of Vietnam's cultural and historical attractions, local cuisine, perceived price, natural environment, entertainment and recreation activities, and destination image on international tourist loyalty. This finding contributes to enhance the consciousness of the tourism sector about the impact these destination attributes have on the sustainable development of HCM City tourism. The sector should pay high attention to protecting the natural environment, especially preserving Can Gio Mangrove forest, which has the potential for eco-tourism. In recent years, due to natural causes mangroves have been dying in many places, and the government should encourage locals to plant mangroves in the Can Gio forest to reduce saltwater intrusion in the agricultural production areas. Further, the forest acts as an air pollution filter. Furthermore, conserving cultural and historical attractions is very important, as they are also major components to consider in the development of sustainable tourism. The government should emphasize the preservation and renovation of cultural and historical attractions in the development of sustainable tourism to increase foreign tourists' return to HCM City. In past years, the tourism in HCM City has encouraged tourist destinations to donate some fees for reconstructing and conserving cultural and historical attractions, such as Notre Dame Cathedral, which is currently being restored.

In addition, travel cost is also an important element when people travel to a foreign country, especially limited-budget tourists. In fact, travelling to HCM City is relatively more expensive than other cities in the region. This makes HCM City less competitive than other destinations. Therefore, the authorities should enact more effective laws and legislations, and take practical actions in order to control and stabilize the market price and protect both domestic and international tourists.

Another remarkable concern is that the destination image of HCM City strongly influenced tourists' perceived value. McDowall [79] conducted a similar study to examine the effects of international tourist satisfaction and destination loyalty in Bangkok, Thailand. The findings were that the success of Bangkok as an attractive destination depends on tourist satisfaction. Other research in Malaysia claims that destination image is the antecedent to tourist satisfaction, which in turn has an effect on destination loyalty [80]. Malaysia was perceived by international tourists as offering natural scenic beauty, especially its beaches, and good facilities such as providing good quality restaurants and hotels.

The study proposes that natural scenic beauty is the competitive advantage of Malaysia, and should be highlighted in promoting Malaysia as a tourist destination.

Besides, regarding the destination image of HCM City, it has many shops, relaxing activities such as spas, water puppets, and Ao Dai shows. In tourists' minds it is still poor and defective compared to our neighboring countries such as Thailand and Malaysia. Mentioning Vietnam, foreigners almost initially think about a small country full of motorcycles rather than a beautiful country with plentiful natural and historical attractions and rich cultural traditions. Although motorcycles have become a characteristic piece of Vietnamese culture, nobody can deny that this destination image negatively affects tourists' perception and their behavior intention, rather than positively. This study's empirical results have also proved this ugly reality.

Thereby, developing tourism is not only designing efficient marketing strategies and tools about a perfect destination image to attract more potential visitors, but also practical actions. Vietnamese tourism should invest in specific strategies and plans in both the short-term and long-term, to comprehensively develop our tourism industry. If we just focus on introducing and advertising about the "timeless charm" of Vietnam with relevant information and pretty images, but the real "timeless charm" is totally different than visitors' expectations, they will not be satisfied with this destination, not come back again, and will not be willing to recommend this place to their families/friends. The core of tourist loyalty comes from what they see and feel rather than what they are told via advertisements and social media.

At present, tourism is a major component of the economy in many countries and regions. This "smokeless industry" with its huge revenue has created plentiful jobs and income for locals. Vietnam also follows this advancing trend and has now become new a destination in Asia and the Pacific region. However, developing tourism needs to focus on the principle of respecting and protecting nature, history, culture and traditional values and national identity, for the sustainable development of tourism.

In addition, our empirical findings rejected the causal relationships of perceived service quality and tourist loyalty toward Ho Chi Minh City. This reflected the poor service quality of almost all Vietnam tourism organizations at present. To rectify this weakness, it is necessary to increase service quality through manpower improvements in the tourism sector. Tourism staff need to improve not only knowledge and language, but must also gain flexible problem-solving skills to adapt to the needs of international tourists. Furthermore, tour operators need to provide various vacation packages, fruitful destination programs, and activities to attract tourists. Travel motivation from the international visitors should also be considered when travel companies design the destination programs. Tour providers should also consider travelers' perception by reducing negative attributes and offering more unforgettable destination experiences. Policy makers play a key role in developing and shaping the country's tourism industry. To manage effectively, it is urgent to control and limit negative attributes to improve tourists' perception and loyalty toward Ho Chi Minh City.

This study has some limitations. First, personal characteristics might highly influence tourist loyalty. These potential moderators need to be added into the model in advance. Second, due to the limitations of time, finance, location, and human resources, the researchers could only deliver the surveys within Ho Chi Minh City. Therefore, the results may change slightly in different areas of the country. Another limitation is that there are many factors affecting tourist loyalty, not only destination attributes, perceived service quality, and perceived value. Hence, it would better strengthen the results by studying a general picture by supplementing other factors into the model. It is recommended that further research should invest more time, human resources, and money to build a more extensive model, which makes the findings more reliable and able to be applied in reality.

## 7. Conclusions

This study identified destination attributes of Ho Chi Minh City, Vietnam and investigated the structural relationships among these destination attributes with tourists' perceived service quality

and perceived value, and their loyalty toward Ho Chi Minh City. The conceptual framework was adapted from relevant theoretical background and related studies. The results provide strong evidence to tourism administrators regarding the role of destination attributes toward tourist loyalty as follows entertainment and activities, the natural environment, the minimization of negative attributes, and safety and security. In terms of the mediating effect of tourist perceived value on destination loyalty, it is important to consider the two major predictors of cultural and historical attractions and destination image. Regarding the indirect effect of perceived service quality, administrators should put high consideration on the two major factors of infrastructure and local cuisine. The results contribute to the literature on tourist destination loyalty and provide a valuable source for administrators in implementing sustainable strategies not only to attract more potential travelers, but also to improve their perception and encourage them to revisit destinations in advance.

**Author Contributions:** K.N.M. proposed the research framework and, together with P.T.M.N., contributed to data collection; P.N.D.N. and P.T.M.N. analyzed the data and wrote the article; K.N.M. contributed to writing and revising article.

**Funding:** This research received granted funding from the Ho Chi Minh City Department of Science and Technology and Department of Tourism of Ho Chi Minh City.

**Conflicts of Interest:** The authors declare no conflicts of interest.

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
