# Peer review of "International Tourists’ Loyalty to Ho Chi Minh City Destination—A Mediation Analysis of Perceived Service Quality and Perceived Value"

_sustainability, doi:10.3390/su11195447_

Round 1

Reviewer 1 Report

The manuscript is interesting, however it needs some improvements before considering it for publishing.

1) Introduction / literature review - well written and easy to follow

2) Conceptual framework – the nine factors should be listed here along with the acronyms in text and/or figure for a better understanding

3) Methdology:

- the year of data collection is missing

- no infomation about the city was presented (location, size, tourism activity etc.) – such information is very important when drawing the final conclusion from your research as one cannon generalise the results...

- Table 2 and Table 3 should be reported in text along with the explanation

4) Discussion: This sections should be revised. In the present form has many phrases that are not supported by any facts-references and are just simple personal opinions (for example:  „Another remarkable concern is that the destination image of Vietnam destination in customers’ mind is still poor and defective compared to our neighbor countries such as Thailand, Malaysia, Indonesia, or even Cambodia." ... and there are more). I am not saying it is not true, but it seems more like a personal opinion. One idea can be to add some stats on some tourism indicators to support what you are saying. 

The study being conducted in only one city, the statement “this study provided practical evidences on the complex relationships between perceived value and tourist loyalty toward Vietnam of foreign tourists” is not supported either. The results of the presented research refer to Ho Chi Minh City and not to the entire country. 

Author Response

Thank you very much for your fruitful comments. We revised point-by-point as your suggestions. Please see the attachment.

Reviewer 2 Report

Dear authors:

Congratulations for the work done. The paper presents a solid and very well structured research. The results are correct and the conclusions appropriate

My only doubt about this research lies in its relevance. Scientifically, it is correct but it is only based on a case study: Hi Chi Minh. Is it representative enough? Could you justify its relevance to be published in a journal of this nature?

On the other hand, I suggest two elements to improve the paper: 

1.Expand the information about the evolution of tourism in the city in the recent years in order to contextualize the city's tourism. The incorporation of graphic material is suggested (chart, figures, etc.).

2. Incorporate a photograph that shows some tourist resource of the city.

Author Response

(The authors gave the same response as above.)

Reviewer 3 Report

This is a very interesting, informative, and internationally important paper with strong methodological and conceptual background, and it deserves publication in Sustainability. I recommend its acceptance after minor improvements. My recommendations are given below.

This paper focuses on Ho Chi Minh City, and this fact should be reflected in the title. This paper is submitted to Sustainability, and, thus, the relevance of this study to sustainable development should be stated clearly in Abstract and Introduction. Avoid phrases like "[17] found...' (page 3). Give author(s) name(s) in such cases. Please, check spelling of Vietnamese names – e.g., Ho Chi Minh on page 1 and Hochiminh on page 6. Be consistent, please! Limitations should be considered in Discussion, not Conclusions. Discussion should also include brief comparison to the results of some other, somewhat similar studies in the other cities of SE Asia or the rest of the world. Something is told already about Thailand and Cambodia, but more details are necessary, as well as enough quantity of references. This paper needs a brief section describing principal tourist attractions and tourist activities in the studied city.

Good luck with revision!

Author Response

(The authors gave the same response as above.)

Round 2

Reviewer 1 Report

The manuscript was improved. Congratulation!

Author Response

Reviewer's comment:

The manuscript was improved. Congratulation!

Authors' response:

Thank you very much for your fruitful feedback.

Reviewer 2 Report

Congratulations for the work done.

Author Response

Reviewer 2 comment:

Congratulations for the work done.

Authors' response:

Thank you very much for your constructive feedback.

This manuscript is a resubmission of an earlier submission. The following is a list of the peer review reports and author responses from that submission.